# Parameter Interval Uncertainty Analysis of Internal Resonance of Rotating Porous Shaft–Disk–Blade Assemblies Reinforced by Graphene Nanoplatelets

**DOI:** 10.3390/ma14175033

**Published:** 2021-09-03

**Authors:** Yi Cai, Zi-Feng Liu, Tian-Yu Zhao, Jie Yang

**Affiliations:** 1School of Control Engineering, Northeastern University at Qinhuangdao, Qinhuangdao 066004, China; caiyi@qhd.neu.edu.cn; 2AVIC Xi’an Aircraft Industry Group Company Ltd., Xi’an 710089, China; lzf2429499849@163.com; 3School of Science, Northeastern University, Shenyang 110819, China; 4School of Engineering, RMIT University, P.O. Box 71, Bundoora, Melbourne, VIC 3083, Australia

**Keywords:** shaft–disk–blade assembly, Chebyshev polynomial approximation method, interval uncertainty, graphene nanoplatelets, porosity

## Abstract

This paper conducts a parameter interval uncertainty analysis of the internal resonance of a rotating porous shaft–disk–blade assembly reinforced by graphene nanoplatelets (GPLs). The nanocomposite rotating assembly is considered to be composed of a porous metal matrix and graphene nanoplatelet (GPL) reinforcement material. Effective material properties are obtained by using the rule of mixture and the Halpin–Tsai micromechanical model. The modeling and internal resonance analysis of a rotating shaft–disk–blade assembly are carried out based on the finite element method. Moreover, based on the Chebyshev polynomial approximation method, the parameter interval uncertainty analysis of the rotating assembly is conducted. The effects of the uncertainties of the GPL length-to-width ratio, porosity coefficient and GPL length-to-thickness ratio are investigated in detail. The present analysis procedure can give an interval estimation of the vibration behavior of porous shaft–disk–blade rotors reinforced with graphene nanoplatelets (GPLs).

## 1. Introduction

Shaft–disk–blade assemblies are commonly applied in many rotor structures, such as gas turbines, aero-engines, and so on. It is reported that more than 60% of faults of shaft–disk–blade assemblies are due to vibration faults. Thus, many scholars have focused on the vibration behaviors of shaft–disk–blade assemblies [1,2,3]. However, modern rotating machinery is faced with high temperatures and high pressures under multiple physical fields. Traditional materials cannot meet the requirements of high strength and light weight at the same time. It is quite necessary to introduce an advanced composite to solve this issue.

Graphene [4,5] is the most popular advanced material in the world because of its outstanding mechanical performance since being discovered. In recent years, GPLs, which have great reinforcing effect at low contents [6], have attracted a lot of attention [7,8,9,10]. Jie Yang, Sritawat Kitipornchai and their partners have contributed many achievements about the vibration characteristics of structures reinforced by GPLs [11,12,13,14,15]. Twinkle et al. [16] studied the vibrations of porous cylindrical panels reinforced with GPLs. Considering the effect of the elastic medium, Mohammad et al. [17] investigated the nonlinear performance of a GPL-reinforced functionally graded (FG) conical panel. Within the frame of the shear deformable theory, Salehi et al. [18] developed an analytical method to obtain the nonlinear vibration behavior of an imperfect porous cylindrical shell reinforced by GPLs. Considering conveying fluid flow, dynamic behaviors of porous sandwich pipes reinforced by GPLs are presented by Nejadi et al. [19]. The analysis of the vibrations of an FG spherical shell reinforced by GPLs was carried out by Liu et al. [20]. Zhao et al. [21,22,23,24,25] conducted a dynamic analysis of FG-GPL-reinforced nanocomposite disk–shaft, blade–disk and blade–shaft rotor systems. By adopting the spectral Chebyshev approach, Mirmeysam et al. [26] studied the inherent characteristics of an FG plate reinforced by GPLs. To the best of the authors’ knowledge, few studies have focused on GPL-reinforced shaft–disk–blade assemblies. Thus, it is important to study the vibration behavior of GPL-reinforced shaft–disk–blade assemblies. In addition, the matrix should be porous metal [27,28,29,30,31,32] to achieve a light weight.

As GPLs are nanofillers, the dimensions of a single GPL are difficult to obtain. Thus, statistical values for the dimensions are adopted. Moreover, the porosity is an approximate measurement for the dimension and density of the pores. The uncertain issue has to exist in the vibration analysis of the porous shaft–disk–blade reinforced by GPLs. If those uncertainties are ignored, the mechanical performance of the porous shaft–disk–blade rotor is partial and cannot be used in practical engineering.

However, the uncertainty analysis of porous structures reinforced with GPLs is extremely limited. Only Baghlani et al. [33] examined the influences of property uncertainties on the free vibration behavior of an FG-GPL-reinforced porous shell. This paper aims to conduct an uncertainty analysis of the internal resonance of porous shaft–disk–blade assemblies reinforced by GPLs. The modeling and internal resonance analysis of the shaft–disk–blade assembly are presented based on the finite element method. In addition, on the basis of the Chebyshev polynomial approximation method, the parameter interval uncertainty analysis of the assembly is conducted. A parametric study is given to investigate the impact of the uncertain porosity coefficient, uncertain GPL length-to-width ratio and uncertain GPL length-to-thickness ratio.

## 2. Physical Model

A spinning shaft–disk–blade model is given in Figure 1. The connections between the shaft, disk and blades are taken to be ideal, where the supports are considered simply supported. The shaft–disk–blade structure is composed of graphene nanoplatelets and porous copper metal foam. 

Based on the open-cell scheme [34], the Poisson’s ratio, mass density and Young’s modulus are:(1)$$\left\{ \begin{array}{l} \upsilon \left(z\right)=\upsilon^{*}\left(z\right)\\ \rho \left(z\right)=\rho^{*}\left(z\right)e_{n}\\ E\left(z\right)=E^{*}\left(z\right)e_{1} \end{array}\right.$$
in which *e*_n_ and *e*_1_ are the mass density coefficients and porosity coefficients, respectively; and *υ^*^*(*z*), *ρ^*^*(*z*) and *E^*^*(*z*) are the Poisson’s ratio, mass density and Young’s modulus in the case of no pores, respectively.

According to the mechanical properties of porous metal foam [35], one can obtain:(2)$$\frac{E\left(z\right)}{E^{*}\left(z\right)}={\left[\frac{\rho \left(z\right)}{\rho^{*}\left(z\right)}\right]}^{2}$$

Thus, one can obtain:(3)$$e_{n}=\sqrt{e_{1}}$$

On the basis of the Halpin–Tsai theory [36], Young’s modulus in the case of no pores is:(4)$$E^{*}\left(z\right)=\frac{5}{8}\left(\frac{1+\xi_{w}\eta_{w}V_{\mathrm{GPL}}}{1-\eta_{w}V_{\mathrm{GPL}}}\right)E_{M}+\frac{3}{8}\left(\frac{1+\xi_{l}\eta_{l}V_{\mathrm{GPL}}}{1-\eta_{l}V_{\mathrm{GPL}}}\right)E_{M}$$
where *ξ**_l_* and *ξ**_w_* are the dimension factors of GPLs, and *ξ**_l_* and *ξ**_w_* are:(5)$$\left\{ \begin{array}{l} \eta_{w}=\frac{1-E_{M}/E_{\mathrm{GPL}}}{1+E_{M}\xi_{w}/E_{\mathrm{GPL}}}\\ \eta_{l}=\frac{1-E_{M}/E_{\mathrm{GPL}}}{1+E_{M}\xi_{l}/E_{\mathrm{GPL}}} \end{array}\right.$$
in which *E*_M_ and *E*_GPL_ are the Young’s modulus of the matrix and GPLs, respectively.

Herein, *ξ_w_* and *ξ_l_* are given by:(6)$$\left\{ \begin{array}{l} \xi_{w}=2w_{\mathrm{GPL}}/h_{\mathrm{GPL}}\\ \xi_{l}=2l_{\mathrm{GPL}}/h_{\mathrm{GPL}} \end{array}\right.$$
where *h*_GPL_, *w*_GPL_ and *l*_GPL_ are the GPL thickness, GPL width and GPL length, respectively.

According to the rule of mixture, *υ^*^*(*z*) and *ρ^*^*(*z*) are given by:(7)$$\left\{ \begin{array}{l} \upsilon^{*}\left(z\right)=\left(\upsilon_{\mathrm{GPL}}-\upsilon_{M}\right)V_{\mathrm{GPL}}+\upsilon_{M}\\ \rho^{*}\left(z\right)=\left(\rho_{\mathrm{GPL}}-\rho_{M}\right)V_{\mathrm{GPL}}+\rho_{M} \end{array}\right.$$
in which *υ*_M_ and *ρ*_M_ are the Poisson’s ratio and mass density of the matrix, respectively; *υ*_GPL_ and *ρ*_GPL_ are the Poisson’s ratio and mass density of GPLs, respectively; and the GPL volume fraction is:(8)$$V_{\mathrm{GPL}}\left(z\right)=\frac{W_{\mathrm{GPL}}\left(z\right)\rho_{M}}{W_{\mathrm{GPL}}\left(z\right)\rho_{M}+\left[1-W_{\mathrm{GPL}}\left(z\right)\right]\rho_{\mathrm{GPL}}}$$

The GPL weight fraction (*W*_GPL_) is expressed as:(9)$$W_{\mathrm{GPL}}\left(z\right)=\lambda W_{0}$$
where *W*_0_ and *λ* are the characteristic value and weight fraction index of GPLs, respectively.

## 3. Finite Element Implementation

Solid elements with eight nodes are adopted in this paper and their displacements are given by: (10)$${\left\{\begin{array}{ccc} u & v & w \end{array}\right\}}^{T}=\sum_{j=1}^{8}N_{j}{\left\{\begin{array}{ccc} u_{j} & v_{j} & w_{j} \end{array}\right\}}^{T}$$
in which (*u_i_*, *v_i_*, *w_i_*) and *N_i_* are the node displacements and shape functions, respectively. 

The expressions of shape functions are:(11)$$\left\{ \begin{array}{l} N_{1}=0.125\left(1-z^{*}\right)\left(1-y^{*}\right)\left(1-x^{*}\right)\\ N_{2}=0.125\left(1-z^{*}\right)\left(1+y^{*}\right)\left(1-x^{*}\right)\\ N_{3}=0.125\left(1+z^{*}\right)\left(1+y^{*}\right)\left(1-x^{*}\right)\\ N_{4}=0.125\left(1+z^{*}\right)\left(1-y^{*}\right)\left(1-x^{*}\right)\\ N_{5}=0.125\left(1-z^{*}\right)\left(1-y^{*}\right)\left(1+x^{*}\right)\\ N_{6}=0.125\left(1-z^{*}\right)\left(1+y^{*}\right)\left(1+x^{*}\right)\\ N_{7}=0.125\left(1+z^{*}\right)\left(1+y^{*}\right)\left(1+x^{*}\right)\\ N_{8}=0.125\left(1+z^{*}\right)\left(1-y^{*}\right)\left(1+x^{*}\right) \end{array}\right.$$
where *z*^*^, *y*^*^ and *x*^*^ are the element coordinates. 

The physical equation is:(12)$$\begin{array}{l} \delta ={\left\{\begin{array}{cccc} \begin{array}{cc} \begin{array}{cc} \delta_{x} & \delta_{y} \end{array} & \delta_{z} \end{array} & \tau_{xy} & \tau_{yz} & \tau_{zx} \end{array}\right\}}^{T}\\ =D{\left\{\begin{array}{cccc} \begin{array}{cc} \begin{array}{cc} \varepsilon_{x} & \varepsilon_{y} \end{array} & \varepsilon_{z} \end{array} & \gamma_{xy} & \gamma_{yz} & \gamma_{zx} \end{array}\right\}}^{T}=D\varepsilon \end{array}$$
where (*γ_xy_*, *γ_yz_*, *γ_zx_*) are the shear strains; (*ε_x_*, *ε_y_*, *ε_z_*) are the normal strains; (*τ_xy_*, *τ_yz_*, *τ_zx_*) are the shear stresses; (*δ_x_*, *δ_y_*, *δ_z_*) are the normal stresses; and the elastic material matrix **D** is:(13)$$D=\frac{E}{1+\upsilon }\left[\begin{array}{cccccc} \frac{1-\upsilon }{1-2\upsilon } & \frac{\upsilon }{1-2\upsilon } & \frac{\upsilon }{1-2\upsilon } & 0 & 0 & 0\\ \frac{\upsilon }{1-2\upsilon } & \frac{1-\upsilon }{1-2\upsilon } & \frac{\upsilon }{1-2\upsilon } & 0 & 0 & 0\\ \frac{\upsilon }{1-2\upsilon } & \frac{\upsilon }{1-2\upsilon } & \frac{1-\upsilon }{1-2\upsilon } & 0 & 0 & 0\\ 0 & 0 & 0 & \frac{1}{2} & 0 & 0\\ 0 & 0 & 0 & 0 & \frac{1}{2} & 0\\ 0 & 0 & 0 & 0 & 0 & \frac{1}{2} \end{array}\right]$$

Then, one can obtain that: (14)$$\begin{array}{l} \varepsilon =\left\{\begin{array}{cccc} \begin{array}{ccc} \frac{\partial u}{\partial x} & \frac{\partial u}{\partial y} & \frac{\partial w}{\partial z} \end{array} & \frac{\partial u}{\partial y}+\frac{\partial v}{\partial x} & \frac{\partial v}{\partial z}+\frac{\partial u}{\partial y} & \frac{\partial u}{\partial z}+\frac{\partial w}{\partial x} \end{array}\right\}\\ \;={\left[\begin{array}{ccc} \begin{array}{cccc} \frac{\partial }{\partial x} & 0 & 0 & \frac{\partial }{\partial y} \end{array} & 0 & \frac{\partial }{\partial z}\\ \begin{array}{cccc} 0 & \frac{\partial }{\partial y} & 0 & \frac{\partial }{\partial x} \end{array} & \frac{\partial }{\partial z} & 0\\ \begin{array}{cccc} 0 & 0 & \frac{\partial }{\partial z} & 0 \end{array} & \frac{\partial }{\partial y} & \frac{\partial }{\partial x} \end{array}\right]}^{T}\left\{\begin{array}{ccc} u & v & w \end{array}\right\} \end{array}$$

Setting
(15)$$\phi ={\left\{u\;v\;w\;\right\}}^{T}={\left\{\varphi_{u}\;\varphi_{v}\;\varphi_{w}\;\right\}}^{T}$$
and supplying Equation (15) into Equation (14) gives:(16)ε=Bϕ
where **B** is the geometric matrix; and **φ***_w_*, **φ***_v_* and **φ***_u_* are the node displacement vectors along the *z*-axis, *y*-axis and *x*-axis, respectively. 

**B** can be written as:(17)$$B={\left[\begin{array}{ccc} \begin{array}{cccc} \frac{\partial }{\partial x} & 0 & 0 & \frac{\partial }{\partial y} \end{array} & 0 & \frac{\partial }{\partial z}\\ \begin{array}{cccc} 0 & \frac{\partial }{\partial y} & 0 & \frac{\partial }{\partial x} \end{array} & \frac{\partial }{\partial z} & 0\\ \begin{array}{cccc} 0 & 0 & \frac{\partial }{\partial z} & 0 \end{array} & \frac{\partial }{\partial y} & \frac{\partial }{\partial x} \end{array}\right]}^{T}\left[\begin{array}{c} N \end{array}\begin{array}{c} N \end{array}\begin{array}{c} N \end{array}\right]$$
in which **N** = {*N*_1_, *N*_2_, …, *N*_8_}.

The deformation energy and kinetic energy are:(18)$$\left\{ \begin{array}{l} U=\int_{V}\frac{1}{2}\delta^{T}\varepsilon dV=\int_{V}\frac{1}{2}\phi^{T}B^{T}DB\phi dV\\ T=\rho \int_{V}\frac{1}{2}\delta^{T}N^{T}N\delta dV \end{array}\right.$$

Considering the expressions:(19)$$\left\{ \begin{array}{l} \frac{\partial U}{\partial \phi }=K\phi \\ \frac{\partial T}{\partial \phi }=M\phi \end{array}\right.$$
give the stiffness matrix **K** and mass matrix **M**, indicated as:(20)$$\left\{ \begin{array}{l} K=\int_{V}B^{T}DBdV\\ M=\rho \int_{V}N^{T}NdV \end{array}\right.$$

The equations of motion are:(21)Mü+Ku=F
where **F** is the harmonic exciting force vector; and **u** is the harmonic response vector. Their expressions are:(22)$$\left\{ \begin{array}{l} F=F_{\max}e^{i\omega t}\\ u=u_{\max}e^{i\omega t} \end{array}\right.$$

Substituting Equation (22) into Equation (21) gives:(23)$$\left(-\omega^{2}M+K\right)u_{\max}=F_{\max}$$

Thus, the relation between the response and frequency can be obtained from Equation (23).

## 4. Interval Uncertainty Analysis

The first kind of Chebyshev orthogonal polynomial is:(24)$$T_{n}\left(x\right)=\cos\left[n\arccos\left(x\right)\right],\;x\in \left[-1,1\right]$$

As the first kind of Chebyshev orthogonal polynomial is orthogonal to:(25)$$\rho \left(x\right)=\sqrt{1-x^{2}}$$

Equation (24) can be obtained as:(26)$$\left\{ \begin{array}{l} T_{0}\left(x\right)=1\\ T_{1}\left(x\right)=x\\ T_{n+1}\left(x\right)=2xT_{n}\left(x\right)-T_{n-1}\left(x\right) \end{array}\right.,\;n=1,2,\ldots$$

Based on the Weierstrass theorem, we can always find a polynomial function *g*(*x*) that satisfies:(27)‖g(x)−f(x)‖≤ε
where *ε* is a small positive real number; and *f*(*x*) is a real function, which is defined in the real interval [−1, 1]. 

In the subspace *T_n_* = span {*T*_0_, *T*_1_, …, *T_n_*}, *f*(*x*) is established as:(28)$$f\left(x\right)\approx g_{n}\left(x\right)=\frac{A_{0}}{2}+\sum_{i=0}^{n}A_{i}T_{i}\left(x\right),\;x\in \left[-1,1\right]$$
in which *A_i_* are undetermined coefficients in the form of:(29)$$\begin{array}{l} A_{0}=\frac{2}{\pi }\int_{-1}^{1}\frac{f\left(x\right)}{\sqrt{1-x^{2}}}dx\approx \sum_{k=1}^{q}{A^{\prime }}_{k}\bar{f}\left(x_{k}\right)\\ A_{i}=\frac{2}{\pi }\int_{-1}^{1}\frac{f\left(x\right)T_{i}\left(x\right)}{\sqrt{1-x^{2}}}dx\approx \sum_{k=1}^{q}{A^{\prime }}_{k}\bar{f}\left(x_{k}\right)T_{i}\left(x_{k}\right) \end{array}$$
where *k* = 1, 2, …, *q*; *q* is the number of interpolation points; and *A*′*_k_* is the Gaussian integral coefficient. It is worth noting that the number of interpolation points must be greater than the order of the approximation equation.

For Chebyshev Gaussian integrals, the interpolation points can be given by:(30)$$x_{k}=\cos\left(\frac{2k-1}{2q}\pi \right),\;k=1,2,\ldots ,q$$

The Gaussian integral coefficients can be calculated by:(31)$${\bar{A}}_{k}=\int_{-1}^{1}\frac{T_{q}\left(x\right)}{\sqrt{1-x^{2}\left(x-x_{k}\right){T^{\prime }}_{q}\left(x_{k}\right)}}dx=\frac{\pi }{q}$$

Substituting Equations (29)–(31) into Equation (28) gives:(32)$$g_{n}\left(x\right)=\frac{1}{q}\sum_{k=1}^{q}\bar{f}\left(x_{k}\right)+\frac{2}{q}\sum_{i=1}^{n}\sum_{k=1}^{q}\bar{f}\left(x_{k}\right)T_{i}\left(x_{k}\right)T_{i}\left(x\right)$$

Thus, the orthogonal approximation is obtained as Equation (32).

In this paper, interval uncertainty analysis is conducted for the porous shaft–disk–blade assembly reinforced by GPLs. The uncertainty parameters are taken into account as the porosity coefficient, GPL length-to-thickness ratio and GPL length-to-width ratio. For convenience, the considered uncertainty parameters are defined as *a^I^*. Its expression is:(33)$$a^{I}=\left[\underline{a},\bar{a}\right]=\left\{a\in R|\underline{a}\leq a\leq \bar{a}\right\}$$
where $\underline{a}$ and $\bar{a}$ are lower boundaries and upper boundaries of the uncertainty parameter and *R* is the real number collection. 

Setting
(34)$$a^{c}=\frac{\underline{a}+\bar{a}}{2},\;\beta =\frac{\underline{a}-\bar{a}}{2}$$
and supplying Equation (34) into Equation (33) give: (35)$$a^{I}=\left[\underline{a},\bar{a}\right]=\left[a^{c}-\beta a^{c},a^{c}+\beta a^{c}\right]$$
in which *β* is the fluctuation coefficient and *a^c^* is the median value of the uncertainty parameter. 

If the number of uncertainty parameters of the shaft–disk–blade assembly is *m*, it can be expressed as:(36)$$\begin{array}{l} a^{I}=\left[a_{1}^{I},a_{2}^{I},\ldots ,a_{m}^{I}\right]\\ a_{i}^{I}=\left[\underline{a_{i}^{I}},\bar{a_{i}^{I}}\right] \end{array},\;i=1,2,\ldots m$$

As the Chebyshev orthogonal approximation is defined in the standard interval [−1, 1], the linear transformation is needed as:(37)$$x_{i}=\frac{2a_{i}-\left({\underline{a}}_{i}+{\bar{a}}_{i}\right)}{{\bar{a}}_{i}-{\underline{a}}_{i}},\;x_{i}\in \left[-1,1\right],\;a_{i}\in \left[{\underline{a}}_{i},{\bar{a}}_{i}\right]$$

For each single uncertainty parameter, the interval steady-state response of the assembly can be expressed as:(38)$$g_{n}\left(x\right)=\frac{1}{q}\sum_{k=1}^{q}\bar{U}\left(x_{k}\right)+\frac{2}{q}\sum_{i=1}^{n}\sum_{k=1}^{q}\bar{U}\left(x_{k}\right)T_{i}\left(x_{k}\right)T_{i}\left(x\right)$$
where $\bar{U}\left(x_{k}\right)$ is the deterministic response at point *x_k_*.

The extremum point is:(39)$$\left\{x_{\mathrm{roots}}|g_{n}^{\prime }\left(x_{\mathrm{roots}}\right)=0\right\}$$

Thus, the steady-state response boundary of the assembly in the presence of the uncertain parameter can be obtained by comparing among *g_n_*(−1), *g_n_*(*x*_roots_) and *g_n_*(1). 

*X*_min_ and *X*_max_ are written as the values of independent variables when *g_n_*(*x*) obtains the smallest and largest values, respectively. In the case of *r* interval variables, uncertainty analysis should be conducted for each one. The two marked corresponding independent variables are:(40)$$\begin{array}{l} x_{\min}=\left[x_{\min}^{1},x_{\min}^{2},\ldots ,x_{\min}^{r}\right]\\ x_{\max}=\left[x_{\max}^{1},x_{\max}^{2},\ldots ,x_{\max}^{r}\right] \end{array}$$

Therefore, the actual parameter vectors of the steady-state response boundaries are:(41)$$\begin{array}{l} a_{\min}=a^{c}+\left(\bar{a}-\underline{a}\right)\times x_{\min}\\ a_{\max}=a^{c}+\left(\bar{a}-\underline{a}\right)\times x_{\max} \end{array}$$

The detailed flow is shown in Figure 2.

## 5. Results and Discussions

Before uncertainty analysis with different parameters is conducted, the experimental validation part is presented first. Due to the manufacturing difficulties of a GPL-reinforced porous structure, a metal alloy shaft–disk–blade assembly is proposed for the validation experiment. From reference [37], it can be noted that the model in this paper has sufficient accuracy.

The uncertainty analysis of the porous shaft–disk–blade assembly reinforced by GPLs is conducted in this part. The blade setting angle, blade thickness, blade width and blade length are 15°, 1.5 mm, 18 mm and 20 mm, respectively; the disk thickness and disk diameter are 20 mm and 78 mm, respectively; the shaft diameter and shaft length are 10 mm and 500 mm, respectively; and the material parameters are *ρ*_M_ = 8960 kg/m^3^, *υ*_M_ = 0.34, *E*_M_ = 130 Gpa, *ρ*_GPL_ = 1062.5 kg/m^3^, *υ*_GPL_ = 0.186, *E*_GPL_ = 1050 Gpa, *e*_1_ = 0.1, *l*_GPL_/*w*_GPL_ = 2, *l*_GPL_/*h*_GPL_ = 10 and *g*_GPL_= 1%. 

Figure 3, Figure 4 and Figure 5 plot the amplitude–frequency response (AFR) of the shaft–disk–blade assembly with a single uncertain porosity coefficient, uncertain GPL length-to-width ratio and uncertain GPL length-to-thickness ratio, respectively. It can be found that the upper boundary of vibration amplitude goes up and the lower boundary of vibration amplitude moves down with the increase in the fluctuation coefficient of the uncertain parameters, which tells us that a larger fluctuation coefficient leads to more uncertain results. Meanwhile, the upper boundary and lower boundary are symmetric about the deterministic response. The formants with different fluctuation coefficients do not shift because the structural damping and external excitation have little effect on the natural frequency of the assembly. Moreover, the uncertain parameters (porosity coefficient, GPL length-to-width ratio and GPL length-to-thickness ratio) make the fluctuation of the resonance peak (around 428 Hz) larger and the fluctuation of the non-resonance peak (away from 428 Hz) smaller. This tells us that structural damping has a great effect on the resonance peak, but little effect on the non-resonance peak.

By comparison with Figure 4, Figure 5 and Figure 6, one can see that the fluctuation coefficient of the GPL length-to-thickness ratio has the greatest influence on the amplitude–frequency response, while that of porosity has the lowest impact. This implies that the thickness and surface area of GPLs are important indexes in the manufacturing of shaft–disk–blade assemblies.

Figure 6, Figure 7 and Figure 8 display the AFR of the shaft–disk–blade assembly with double uncertain parameters (porosity coefficient, GPL length-to-thickness ratio and GPL length-to-width ratio). One can see that the fluctuation of the amplitude–frequency response with double uncertain parameters is much stronger than that with a single uncertain parameter. In the case of double uncertain parameters, the width of the resonance peak region is increased significantly. However, the non-resonance peak region is affected very little by the double uncertain parameters. In addition, it is found that the AFR with an uncertain GPL length-to-thickness ratio and GPL length-to-width ratio has the largest fluctuation effect compared to the other two cases. This implies that the dimensions of GPLs have greater effects on the mechanical performance of the rotor system as thinner GPLs with larger surface areas provide better load transfer capability.

## 6. Conclusions

In this paper, a GPL-reinforced porous shaft–disk–blade assembly is established by employing the finite element method. According to the Chebyshev polynomial approximation method, the parameter interval uncertainty analysis of the internal resonance of the shaft–disk–blade assembly is carried out. Some conclusions are drawn as follows:The uncertain parameters make the fluctuation of the resonance peak larger and the fluctuation of the non-resonance peak smaller.The fluctuation coefficient of the GPL length-to-thickness ratio has the greatest influence on the amplitude–frequency response, while that of porosity has the lowest impact.The fluctuation of the amplitude–frequency response with double uncertain parameters is much stronger than that with a single uncertain parameter.The dimensions of GPLs have greater effects on the vibration behavior of the shaft–disk–blade assembly as thinner GPLs with larger surface areas provide better load transfer capability.

## Figures and Tables

**Figure 1 materials-14-05033-f001:**
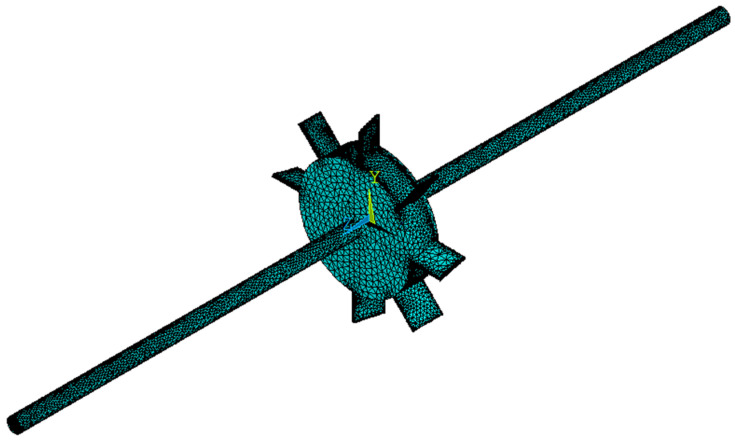
A porous shaft–disk–blade assembly reinforced with GPLs.

**Figure 2 materials-14-05033-f002:**
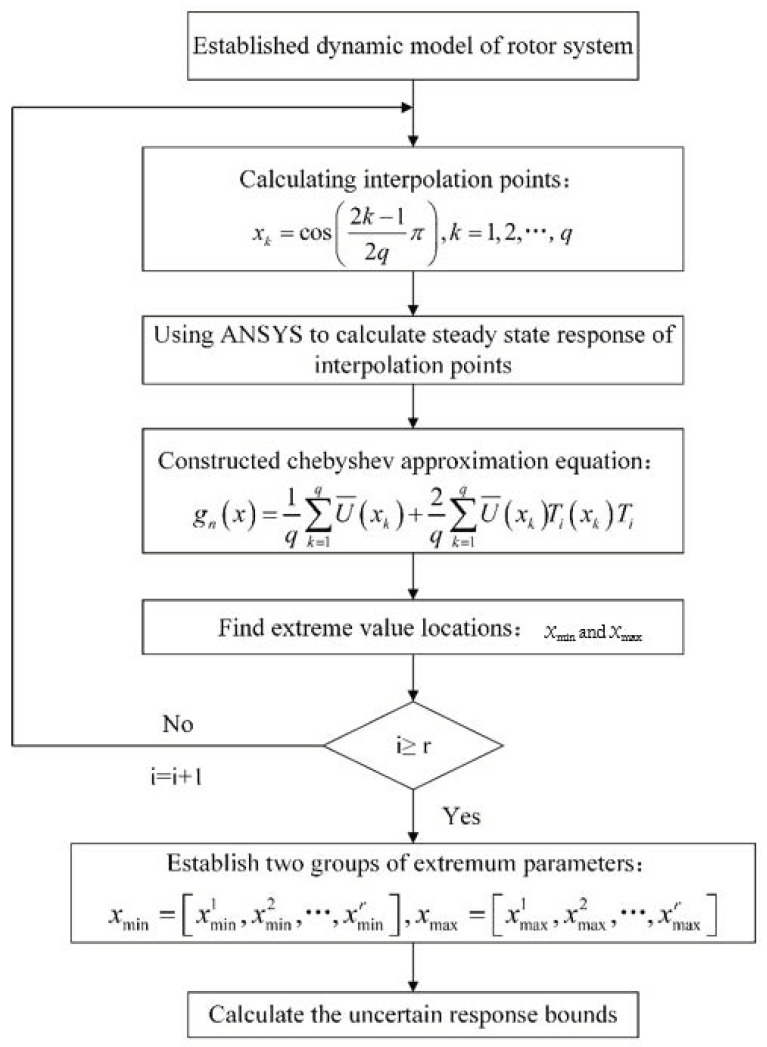
Flow of uncertainty analysis for the porous shaft–disk–blade assembly reinforced by GPLs.

**Figure 3 materials-14-05033-f003:**
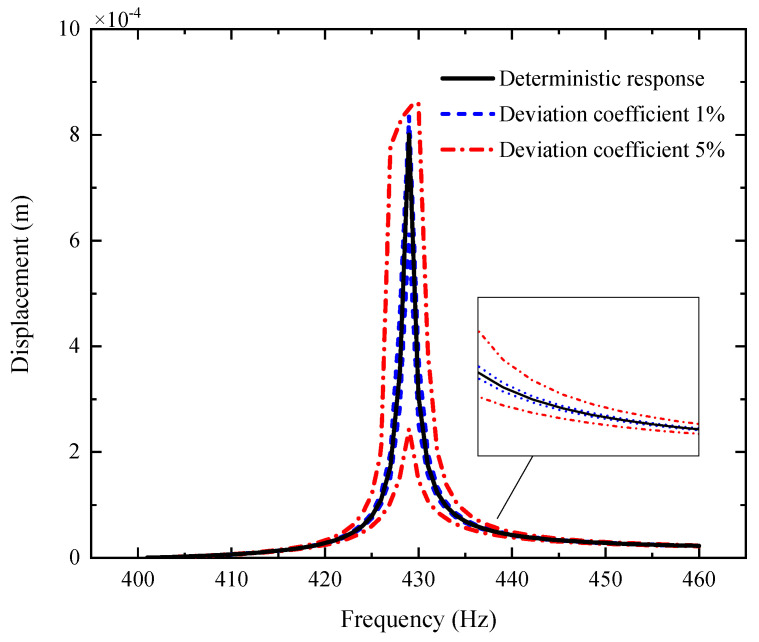
AFR of the shaft–disk–blade assembly with uncertain GPL length-to-thickness ratio.

**Figure 4 materials-14-05033-f004:**
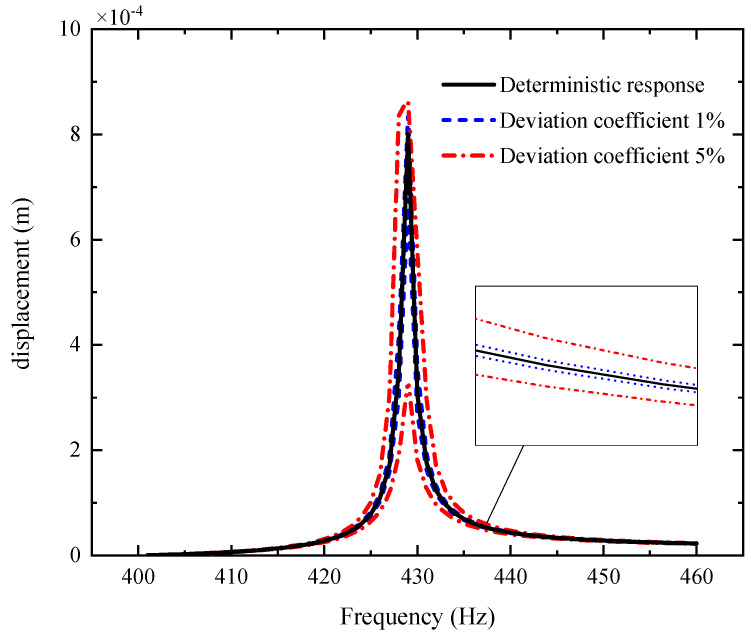
AFR of the shaft–disk–blade assembly with uncertain GPL length-to-width ratio.

**Figure 5 materials-14-05033-f005:**
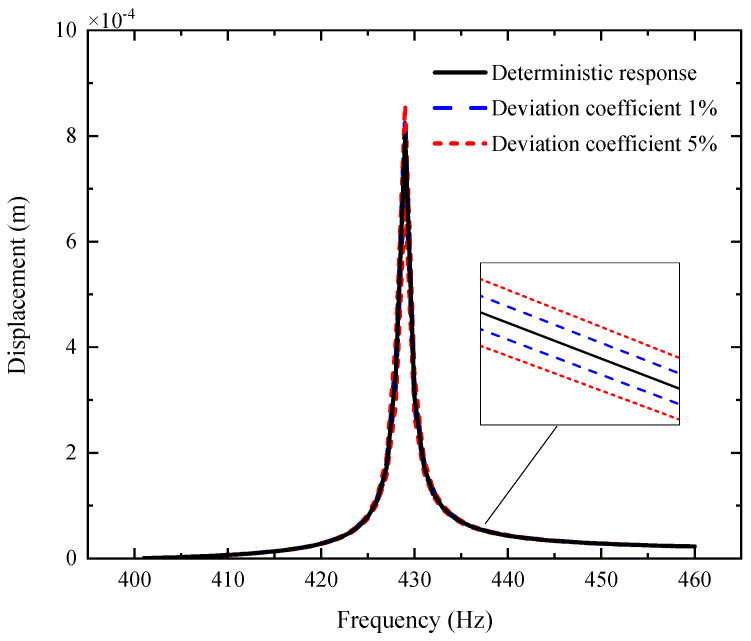
AFR of the shaft–disk–blade assembly with uncertain porosity.

**Figure 6 materials-14-05033-f006:**
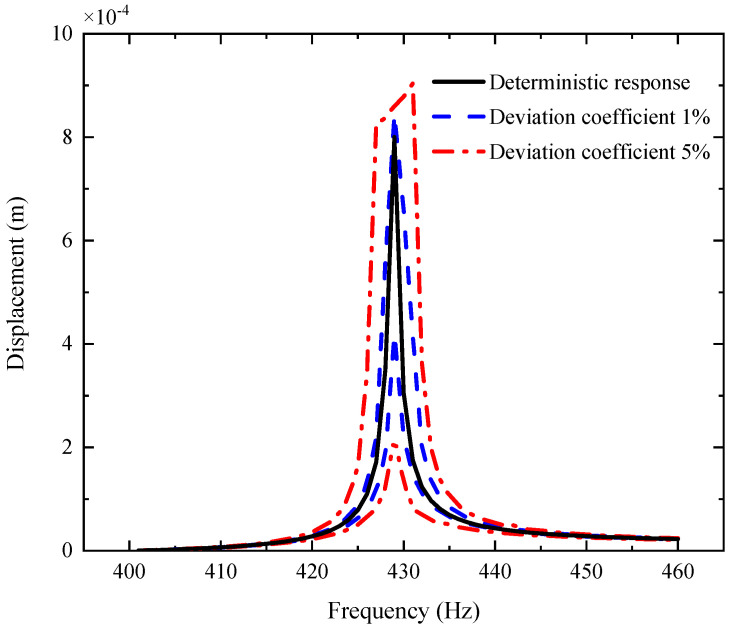
AFR of the shaft–disk–blade assembly with uncertain GPL length-to-width ratio and GPL length-to-thickness ratio.

**Figure 7 materials-14-05033-f007:**
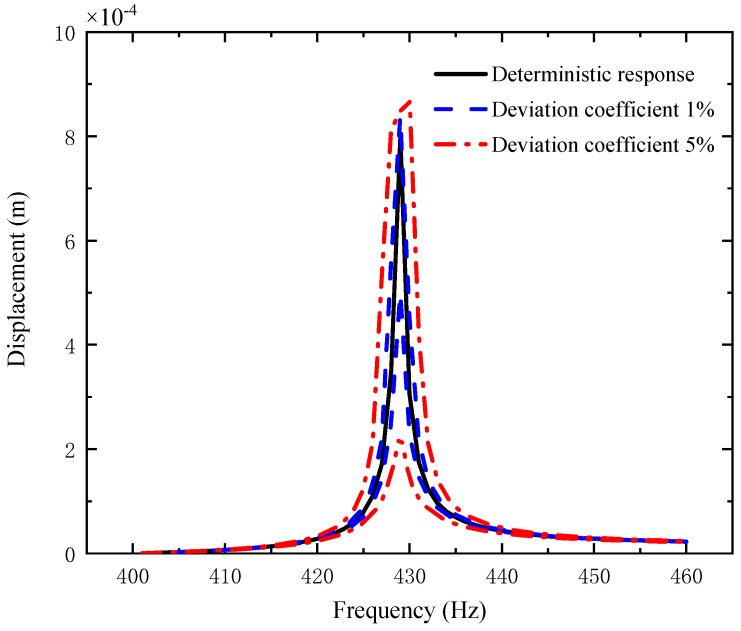
AFR of the shaft–disk–blade assembly with uncertain GPL length-to-thickness ratio and porosity coefficient.

**Figure 8 materials-14-05033-f008:**
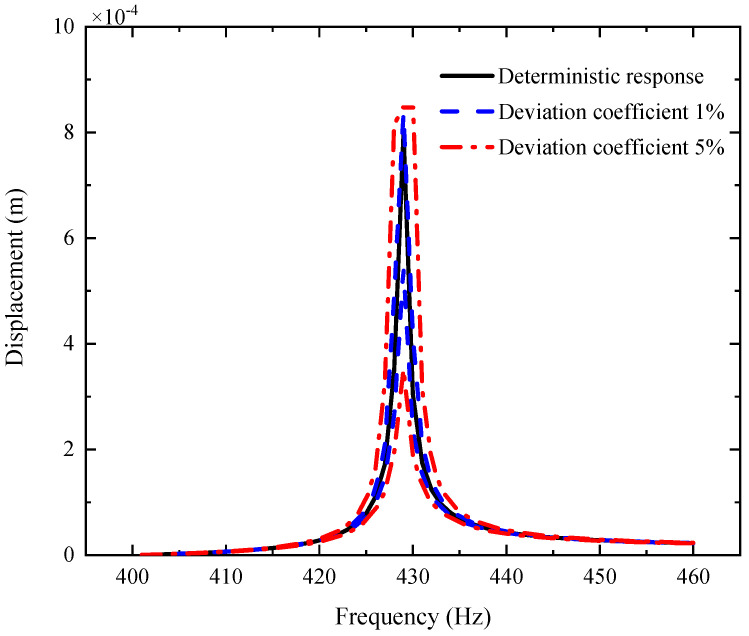
AFR of the shaft–disk–blade assembly with uncertain GPL length-to-width ratio and porosity coefficient.

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
