# Peer review of "Parameter Interval Uncertainty Analysis of Internal Resonance of Rotating Porous Shaft–Disk–Blade Assemblies Reinforced by Graphene Nanoplatelets"

_materials, 2021, doi:10.3390/ma14175033_

Round 1
Reviewer 1 Report
An article has been submitted to Materials journal, however, material aspects are completely ignored in this article.
The novelty aspect is hardly described in comparison with the analysed literature.
The chapter "material property" contains practically no material science aspects. In addition, the material data have not been sufficiently presented, especially when talking about the strut called by the authors "porous shaft-disk-blade assemblies reinforced by graphene nanoplatelets". Complete lack of description in this regard.
Chapter 3 contains known and basic information from FEM, which in no way reflects or harmonizes with the description of the structure, which is called "porous shaft-disk-blade assemblies reinforced by graphene nanoplatelets" and refers to an isotropic material characterized by continuity.
The presented model relates to the dynamic problem, however, the authors completely ignore the problem of damping despite presenting its existence in equation 23.
The presented conclusions sound very engineering without deep thoughts and scientific aspects.
Author Response
Responses to reviewers’ comments
We highly appreciate the reviewers having taken time to read our paper. All comments made by the reviewers have been carefully considered, and the revisions have been made accordingly. The revisions are highlighted in red in the revised manuscript. Point-to-point revisions are listed below for the convenience of the editor and reviewers.
Reviewer #1
Comment 1: An article has been submitted to Materials journal; however, material aspects are completely ignored in this article. The novelty aspect is hardly described in comparison with the analyzed literature. The chapter "material property" contains practically no material science aspects.
Reply: Materials journal has published quite a few research papers on the mechanical behaviors of advanced composite structures. This paper belongs to this category and is submitted to the special issue “Modeling and Analysis of static, Dynamic, and Thermal Behavior of Shell, Plate, and Beam Structures”. The focus and major contribution of the present study are the development of theoretical formulations and in-depth understanding of the parameter interval uncertainty of internal resonance of a rotating porous shaft-disk-blade structure reinforced by graphene nanoplatelets (GPLs) which is an important topic that has not been dealt with in previous studies.
Comment 2: In addition, the material data have not been sufficiently presented, especially when talking about the strut called by the authors "porous shaft-disk-blade assemblies reinforced by graphene nanoplatelets". Complete lack of description in this regard.
Reply: The shaft-disk-blade rotor system is made of porous copper foam as the metal matrix and graphene nanoplatelets as the reinforcing nanofillers. This has been added in the Material property section. Moreover, the material parameters used in this paper are EGPL= 1050Gpa, EM= 130Gpa, uGPL= 0.186, uM= 0.34, rGPL= 1062.5kg/m3, rM= 8960kg/m3, gGPL= 1%, lGPL/hGPL= 10, lGPL/wGPL= 2 and e1= 0.1. These details have been added immediately before Figure 4.
Comment 3: Chapter 3 contains known and basic information from FEM, which in no way reflects or harmonizes with the description of the structure, which is called "porous shaft-disk-blade assemblies reinforced by graphene nanoplatelets" and refers to an isotropic material characterized by continuity.
Reply: The present study considers uniformly distributed GPLs and porosity, which means that each component in the system is homogeneous and isotropic.
Comment 4: The presented model relates to the dynamic problem; however, the authors completely ignore the problem of damping despite presenting its existence in equation 23.
Reply: This paper is focused on the undamped internal resonance characteristics of a graphene nanoplatelets reinforced porous shaft-disk-blade assembly. To be consistent, the damping term is removed from equations (21) and (23). The damped dynamic behavior of such a system will be the topic of our future research.
Comment 5: The presented conclusions sound very engineering without deep thoughts and scientific aspects.
Reply: The Conclusions section has been improved.
Reviewer #2
The manuscript deals with an interesting topic of uncertainty analysis of internal resonance of a rotating porous shaft-disk-blade assembly. Results are very interesting and worth publishing. Reviewer recommends following modification before being accepted for publication.
Comment 1: It was stated that nanocomposite assembly was considered with graphene nanoplatelet and porous metal matrix. However, it is not clear which metal and how were the properties selected. Please explain these.
Reply: The shaft-disk-blade rotor system is made of porous copper foam as the metal matrix and graphene nanoplatelets as the reinforcing nanofillers. This has been added in the Material property section. Moreover, the material parameters used in this paper are EGPL= 1050Gpa, EM= 130Gpa, uGPL= 0.186, uM= 0.34, rGPL= 1062.5kg/m3, rM= 8960kg/m3, gGPL= 1%, lGPL/hGPL= 10, lGPL/wGPL= 2 and e1= 0.1. These details have been added immediately before Figure 4.
Comment 2: It seems everything was linear elastic, please state all the assumptions and limitations with justification in the manuscript.
Reply: It is assumed that the blades, disk and shaft are ideally connected, and the blade is simply supported at both ends. The shaft-disk-blade rotor system is made of porous copper foam as metal matrix and graphene nanoplatelets as reinforcing nanofillers. These assumptions have been mentioned in the paragraph after Figure 1.
Comment 3: Page 2 paragraph 2: Literature review should be expanded more, with discussion on computational modelling of porous (1,2,4) and nanomaterials (3). Following references may be useful along with references there in. 1.A variational void coalescence model for ductile metals, Computational Mechanics 49 (2), 185-195, 2012.
Also, surface defects may cause fatigue failure under vibration (cyclic loading) therefore techniques related to surface defect machining can improve component performance. Following references may be used to enhance literature review portion. Smooth particle hydrodynamics study of surface defect machining for diamond turning of silicon, The International Journal of Advanced Manufacturing Technology 88, 2461-2476, 2017.
Reply: The suggested references have been cited.
Comment 4: page 3 equation (4), all symbols should be explained in the text in a physical sense. For example, what is the meaning of etal_l in equation (4 & 5). Same applies to any other equation in the manuscript.
Reply: The description is added after equation (4) in the revised manuscript.
Comment 5: - line 81: "ofthe" to "of the"?
- line 88: wGPL should be w_GPL(subscript)?
- line 99: wit to with?
- Table 1: Present means FE? may be change it to FE then.
- Table 1: it seems error increases with increasing offset, why is that so?
- Check for grammar and spelling errors using spell checker, if possible
Reply: The language has been carefully checked and the motioned errors have been corrected. As suggested, “Present” in Table 1 has been changed to FE. The bigger error with an increase in offset in Table 1 is due to the increased error in experiments as the offset increases.
Reviewer #3
In this paper entitled "Parameter interval uncertainty analysis of internal resonance of rotating porous shaft-disk-blade assemblies reinforced by graphene nanoplatelets" author reported the parameter interval uncertainty analysis of internal resonance. This paper is very interesting, but I have some minor comments.
Comment 1: There is no date about the prepared GPL reinforced porous blade-disk-shaft assembly. Author needs to provide the photograph of this.
Reply: The shaft-disk-blade rotor system is made of porous copper foam as the metal matrix and graphene nanoplatelets as the reinforcing nanofillers. This has been added in the Material property section. Moreover, the material parameters used in this paper are EGPL= 1050Gpa, EM= 130Gpa, uGPL= 0.186, uM= 0.34, rGPL= 1062.5kg/m3, rM= 8960kg/m3, gGPL= 1%, lGPL/hGPL= 10, lGPL/wGPL= 2 and e1= 0.1. These details have been added immediately before Figure 4.
Comment 2: In addition, this porous structure should be investigated by SEM. I recommend to include this data in revised version.
Reply: Thank you for this comment. We agree that SEM is an important tool to study and characterize the microstructure of material but our paper belongs to the special issue “Modeling and Analysis of static, Dynamic, and Thermal Behavior of Shell, Plate, and Beam Structures” and is focused on the development of theoretical formulations and in-depth understanding of the parameter interval uncertainty of internal resonance behavior of a rotating porous shaft-disk-blade structure reinforced by graphene nanoplatelets (GPLs).
Comment 3: Include some related papers in introduction part.
- Polymers, 12(8), 1666 (2020); Small Methods, 3, 11, 1900277 (2019)
Reply: The motioned references have been cited.

Reviewer 2 Report
The manuscript deals with an interesting topic of uncertainty analysis of internal resonance of a rotating porous shaft-disk-blade assembly. Results are very interesting and worth publishing. Reviewer recommends following modification before being accepted for publication.
- It was stated that nanocomposite assembly was considered with graphene nanoplatelet and porous metal matrix. However, it is not clear which metal and how were the properties selected. Please explain these.
- It seems everything was linear elastic, please state all the assumptions and limitations with justification in the manuscript.
- Page 2 paragraph 2: Literature review should be expanded more, with discussion on computational modelling of porous (1,2,4) and nanomaterials(3). Following references may be useful along with references there in.
1.A variational void coalescence model for ductile metals, Computational Mechanics 49 (2), 185-195, 2012.
- Also, surface defects may cause fatigue failure under vibration (cyclic loading) therefore techniques related to suface defect machining can improve component performance. Following references may be used to enhance literature review portion.
Smooth particle hydrodynamics study of surface defect machining for diamond turning of silicon,
The International Journal of Advanced Manufacturing Technology 88, 2461-2476, 2017.
- page 3 equation (4), all symbols should be explained in the text in a physical sense. For example, what is the meaning of etal_l in equation (4 & 5). Same applies to any other equation in the manuscript.
- line 81: "ofthe" to "of the"?
- line 88: wGPL should be w_GPL(subscript)?
- line 99: wit to with?
- Table 1: Present means FE? may be change it to FE then.
- Table 1: it seems error increases with increasing offset, why is that so?
- Check for grammar and spelling errors using spell checker, if possible.
Author Response

(The authors gave the same response as above.)

Reviewer 3 Report
In this paper entitled "Parameter interval uncertainty analysis of internal resonance of rotating porous shaft-disk-blade assemblies reinforced by graphene nanoplatelets" author reported the parameter interval uncertainty analysis of internal resonance. This paper is very interesting but I have some minor comments.
1) There is no date about the prepared GPL reinforced porous blade-disk-shaft assembly. Author needs to provide the photograph of this.
2) In addition, this porous structure should be investigated by SEM. I recommend to include this data in revised version.
3) Include some related papers in introduction part.
- Polymers, 12(8), 1666 (2020); Small Methods, 3, 11, 1900277 (2019)
Author Response

(The authors gave the same response as above.)

Reviewer 4 Report
In this manuscript, Authors studied the parameter interval uncertainty of the porous shaft-disc-blade assembly's internal resonance powered by GPLs. They showed the shaft-disk-blade assembly's model and internal resonance analysis by using finite element method, and the parameter interval uncertainty analysis of the assembly by the Chebyshev polynomial approximation method. They examined and compared the effects of the single and double uncertain parameters (GPL length-to-thickness ratio, uncertain GPL length-to-width ratio and uncertain porosity coefficient).
They present the open issues in the literature, and their contribution to the literature.
In Line 84: V_GPL and n_1 and n_w are not described in equation (4).
In Line 85: there is a repetition.
In Line 102: Please rewrite the sentence.
In Line 113: T is not described.
In Line 127: What is ü ?
In Line 158: Equation number is not given " Eq. ()."
Equation (35) is missing.
Please check all the equations, descriptions and the references for the equations again.
In Line 214: please describe "FE"
I can advice to add the causes and consequences of the results of the Figures to make better discussion. It is not enough to just read what is seen in the Figures. For instance, the sentences starting in Line 234 can be explained.
"It can be found that the upper boundary of vibration amplitude goes up and lower boundary of vibration amplitude moves down with the increase of the fluctuation coefficient of the uncertain parameters (GPL length-to-thickness ratio, GPL length-to-width ratio and porosity). Meanwhile, the upper boundary and lower boundary are symmetric about the deterministic 237 response."
'What does this result imply' issue can be added for the all figures in the manuscript.
This issue is valid also for the conclusion after the Line 272.
Author Response
Responses to reviewers’ comments
We highly appreciate the reviewers having taken time to read our paper. All comments made by the reviewers have been carefully considered, and the revisions have been made accordingly. The revisions are highlighted in red in the revised manuscript. Point-to-point revisions are listed below for the convenience of the editor and reviewers.
Review 4
In this manuscript, Authors studied the parameter interval uncertainty of the porous shaft-disc-blade assembly's internal resonance powered by GPLs. They showed the shaft-disk-blade assembly's model and internal resonance analysis by using finite element method, and the parameter interval uncertainty analysis of the assembly by the Chebyshev polynomial approximation method. They examined and compared the effects of the single and double uncertain parameters (GPL length-to-thickness ratio, uncertain GPL length-to-width ratio and uncertain porosity coefficient). They present the open issues in the literature, and their contribution to the literature.
In Line 84: V_GPL and n_1 and n_w are not described in equation (4).
Reply: The parameters in Eq. (4) have been described in Eq. (5-8).
In Line 85: there is a repetition.
Reply: The repetition has been deleted.
In Line 102: Please rewrite the sentence.
Reply: The sentences in Line 104 and 106 in the new manuscript are revised.
In Line 113: T is not described.
Reply: The superscript T means transpose.
In Line 127: What is ü?
Reply: ü is the second derivative of u with respect to time t.
In Line 158: Equation number is not given " Eq. ()."
Reply: It has been added as “Eq. (32)”.
Equation (35) is missing.
Reply: Eq. (35) is added.
Please check all the equations, descriptions and the references for the equations again. In Line 214: please describe "FE".
Reply: The “FE” has been described.
I can advise to add the causes and consequences of the results of the Figures to make better discussion. It is not enough to just read what is seen in the Figures. For instance, the sentences starting in Line 234 can be explained. "It can be found that the upper boundary of vibration amplitude goes up and lower boundary of vibration amplitude moves down with the increase of the fluctuation coefficient of the uncertain parameters (GPL length-to-thickness ratio, GPL length-to-width ratio and porosity). Meanwhile, the upper boundary and lower boundary are symmetric about the deterministic 237 responses."
'What does this result imply' issue can be added for all figures in the manuscript. This issue is valid also for the conclusion after the Line 272.
Reply: The descriptions below is added
from Line 243-245:
“This implies that the random errors in the material preparation and fabrication may lead to significantly affected structural responses. The present analysis can give accurate estimation based on the uncertainty of material parameters.”
from Line 272-274:
This implies that the dimensions of GPLs have greater effects on the vibration behavior of the shaft-disk-blade assembly as thinner GPLs with larger surface areas provide better load transfer capability.

Reviewer 5 Report
The authors have presented the parameter interval uncertainty analysis of internal resonance of a rotating porous shaft-disk-blade assembly reinforced by graphene nanoplatelets (GPLs) with the special attention to the uncertainty of GPL length-to-thickness ratio, GPL length-to-width ratio and porosity coefficient. However, the significance of the GPL length-to-thickness ratio, GPL length-to-width ratio and porosity coefficient to the performance of the rotating porous shaft-disk-blade assembly has not been clearly explained.
Therefore, I like to accept this manuscript with major revision. The authors may consider the following points to improve the manuscript:
Major:
- The authors need to include a background of the significance of the GPL length-to-thickness ratio, GPL length-to-width ratio and porosity coefficient to the performance of the rotating porous shaft-disk-blade assembly in the Introduction Section.
- The ‘Materials Property’ section needs to be renamed.
- ‘Conclusion’ may be rewritten to address the significance of the key terms on the performance of the shaft-disk-blade.
Minor:
- “, respectively” should be added at the ending of the sentence in line 210.
- “to” has be added after ‘reached’ in line 211.
- “found to be” has be added after ‘are’ in line 214.
- The 2 sentences in lines 219-221 may be merged to be 1.
- The authors may need to identify the ‘resonance peak’ and ‘non-resonance peak’ in figures 4, 5 and 6.
- The authors may like to discuss about the ‘deviation coefficient’ used in figures 4-9.

Author Response
Responses to reviewers’ comments
We highly appreciate the reviewers having taken time to read our paper. All comments made by the reviewers have been carefully considered, and the revisions have been made accordingly. The revisions are highlighted in red in the revised manuscript. Point-to-point revisions are listed below for the convenience of the editor and reviewers.
Comments to the authors:
The authors have presented the parameter interval uncertainty analysis of internal resonance of a rotating porous shaft-disk-blade assembly reinforced by graphene nanoplatelets (GPLs) with the special attention to the uncertainty of GPL length-to-thickness ratio, GPL length-to-width ratio and porosity coefficient. However, the significance of the GPL length-to-thickness ratio, GPL length-to-width ratio and porosity coefficient to the performance of the rotating porous shaft-disk-blade assembly has not been clearly explained.
Therefore, I like to accept this manuscript with major revision. The authors may consider the following points to improve the manuscript:
Major:
- The authors need to include a background of the significance of the GPL length-to-thickness ratio, GPL length-to-width ratio and porosity coefficient to the performance of the rotating porous shaft-disk-blade assembly in the Introduction Section.
Reply: The introduction section has been revised (As GPLs are nanofillers, the dimensions of single GPL are difficult to obtain. Thus, the statistical values for the dimensions are adopting. Moreover, the porosity is an approximate measurement for the dimension and density of the pores. The uncertain issue has to exist in the vibration analysis of the porous shaft-disk-blade reinforced by GPLs. If those uncertainties are ignored, the mechanical performance of the rotating porous shaft-disk-blade assembly are partial and cannot be used in practical engineering.).
- The ‘Materials Property’ section needs to be renamed.
Reply: It has been renamed (Physical model).
- ‘Conclusion’ may be rewritten to address the significance of the key terms on the performance of the shaft-disk-blade.
Reply: The conclusions have been revised ((1) the uncertain parameters make the fluctuation of the resonance peak larger and the fluctuation of the non-resonance peak smaller. (2) the fluctuation coefficient of GPL length-to-thickness ratio has the greatest influence on the amplitude-frequency response, while that of porosity has the lowest impact. (3) the fluctuation of amplitude-frequency response with double uncertain parameters is much stronger than that with single uncertain parameter. (4) the dimensions of GPLs have greater effects on the vibration behavior of the shaft-disk-blade assembly as thinner GPLs with larger surface areas provide better load transfer capability).
Minor:
- “, respectively” should be added at the ending of the sentence in line 210.
Reply: “, respectively” has been added.
- “to” has be added after ‘reached’ in line 211.
Reply: “to” has been added.
- “found to be” has be added after ‘are’ in line 214.
Reply: “found to be” has been added.
- The 2 sentences in lines 219-221 may be merged to be 1.
Reply: The 2 sentences has be merged to be 1. (It should be mentioned that all the errors are less than 2% for different disk locations, which implies that the modelling and vibration analysis in this paper is accurate enough)
- The authors may need to identify the ‘resonance peak’ and ‘non-resonance peak’ in figures 4, 5 and 6.
Reply: The ‘resonance peak’ and ‘non-resonance peak’ are identified (around 428Hz) and (away from 428Hz), respectively. (Moreover, the uncertain parameters (GPL length-to-thickness ratio, GPL length-to-width ratio and porosity) make the fluctuation of the resonance peak (around 428Hz) larger and the fluctuation of the non-resonance peak (away from 428Hz) smaller.)
- The authors may like to discuss about the ‘deviation coefficient’ used in figures 4-9.
Reply: The ‘deviation coefficient’ has been discussed. (It can be found that the upper boundary of vibration amplitude goes up and lower boundary of vibration amplitude moves down with the increase of the fluctuation coefficient of the uncertain parameters, which tells that larger fluctuation coefficient leads to greater uncertain results.)

Round 2
Reviewer 1 Report
In the paper there is nothing new and original information related to the material part and the main body text is completely out of the journal scope where clearly is stated:
Topics Covered
- Class of materials include ceramics, glasses, polymers (plastics), semiconductors, magnetic materials, medical implant materials and biological materials, silica and carbon materials, metals and metallic alloys. All kinds of functional materials including material for dentistry. Composites, coatings and films, pigments. Classes of materials such as ionic crystals; covalent crystals; metals; intermetallics.
- Materials science or materials engineering. Nanoscience and nanotechnology will be considered also.
- Characterization techniques such as electron microscopy, x-ray diffraction, calorimetry, nuclear microscopy (HEFIB), Rutherford backscattering, neutron diffraction, etc.
- Fundamental research: Condensed matter physics and materials physics. Continuum mechanics and statistics. Mechanics of materials. Tribology (friction, lubrication and wear). Solid-state physics.
Author Response
Responses to reviewers’ comments
We highly appreciate the reviewers having taken time to read our paper. All comments made by the reviewers have been carefully considered, and the revisions have been made accordingly. The revisions are highlighted in red in the revised manuscript. Point-to-point revisions are listed below for the convenience of the editor and reviewers.
Review 1
In the paper there is nothing new and original information related to the material part and the main body text is completely out of the journal scope where clearly is stated:
Topics Covered:
Class of materials include ceramics, glasses, polymers (plastics), semiconductors, magnetic materials, medical implant materials and biological materials, silica and carbon materials, metals and metallic alloys. All kinds of functional materials including material for dentistry. Composites, coatings and films, pigments. Classes of materials such as ionic crystals; covalent crystals; metals; intermetallic.
Materials science or materials engineering. Nanoscience and nanotechnology will be considered also.
Characterization techniques such as electron microscopy, x-ray diffraction, calorimetry, nuclear microscopy (HEFIB), Rutherford backscattering, neutron diffraction, etc.
Fundamental research: Condensed matter physics and materials physics. Continuum mechanics and statistics. Mechanics of materials. Tribology (friction, lubrication and wear). Solid-state physics.
Reply: This paper is submitted to the special issue “Modeling and Analysis of static, Dynamic, and Thermal Behavior of Shell, Plate, and Beam Structures”. Below is the general information of this Special Issue.
This Special Issue deals with analytical and computational methods in the mechanics of beams, plates, and shells. The main areas of interest of this edition include linear and nonlinear models of elasticity and plasticity of these structures; thermoelasticity; problems of vibrations, wave propagation, stability of beams, plates, and shells; heat conductivity; microheterogeneous structures; layered structures; and structures made of materials with special properties-metamaterials, auxetic materials, porous materials, biomaterials, and functionally graded materials.
We invite you to submit your scientific papers on the latest research results in these aspects of the mechanics of beams, plates, and shells, with an emphasis on applications in all areas of mechanics, biomechanics, and civil engineering.
The above information is also available from the link below
https://www.mdpi.com/journal/materials/special_issues/Model_Anal_Stat_Dyn_Therm_Behav_Shell_Plate_Beam_Struct
